PlantCV v2: Image analysis software for high-throughput plant phenotyping

Gehan Malia A. mgehan@danforthcenter.org 1
Fahlgren Noah nfahlgren@danforthcenter.org 1
Abbasi Arash 1
Berry Jeffrey C. 1
Callen Steven T. 1 8
Chavez Leonardo 1
Doust Andrew N. 2
Feldman Max J. 1
Gilbert Kerrigan B. 1
Hodge John G. 2
Hoyer J. Steen 1 3
Lin Andy 1 9
Liu Suxing 4 10
Lizárraga César 1 11
Lorence Argelia 5
Miller Michael 1 12
Platon Eric 6
Tessman Monica 1 2
Sax Tony 7
1 Donald Danforth Plant Science Center , St. Louis , MO , United States of America
2 Department of Plant Biology, Ecology, and Evolution, Oklahoma State University , Stillwater , OK , United States of America
3 Computational and Systems Biology Program, Washington University in St. Louis , St. Louis , MO , United States of America
4 Arkansas Biosciences Institute, Arkansas State University , Jonesboro , AR , United States of America
5 Arkansas Biosciences Institute, Department of Chemistry and Physics, Arkansas State University , Jonesboro , AR , United States of America
6 Cosmos X , Tokyo , Japan
7 Missouri University of Science and Technology , Rolla , MO , United States of America
8 Current affiliation:  Monsanto Company , St. Louis , MO , United States of America
9 Current affiliation:  Unidev , St. Louis , MO , United States of America
10 Current affiliation:  Department of Plant Biology, University of Georgia , Athens , GA , United States of America
11 Current affiliation:  CiBO Technologies , Cambridge , MA , United States of America
12 Current affiliation:  Department of Agronomy and Horticulture, Center for Plant Science Innovation, Beadle Center for Biotechnology, University of Nebraska - Lincoln , Lincoln , NE , United States of America
Loraine Ann
Electronic publication date: 2017 Dec 1
Publication date: 2017
Volume: 5
Electronic Location ID: e4088
Received 2017 Sep 7; Accepted 2017 Nov 3
Copyright: ©2017 Gehan et al.
Copyright year: 2017
Copyright holder: Gehan et al.
License: This is an open access article distributed under the terms of the Creative Commons Attribution License, which permits unrestricted use, distribution, reproduction and adaptation in any medium and for any purpose provided that it is properly attributed. For attribution, the original author(s), title, publication source (PeerJ) and either DOI or URL of the article must be cited.
License URL: https://creativecommons.org/licenses/by/4.0/

Keywords: Plant phenotyping, Image analysis, Computer vision, Machine learning, Morphometrics

Funding: Donald Danforth Plant Science Center US National Science Foundation IIA-1430427 IIA-1430428 IIA-1355406 IOS-1202682 MCB-1330562 DBI-1156581 US Department of Energy DE-AR0000594 DE-SC0014395 US Department of Agriculture MOW-2012-01361 2016-67009-25639 This work was supported by the Donald Danforth Plant Science Center, the US National Science Foundation (IIA-1430427, IIA-1430428, IIA-1355406, IOS-1202682, MCB-1330562, and DBI-1156581), the US Department of Energy (DE-AR0000594, DE-SC0014395), and the US Department of Agriculture (MOW-2012-01361 and 2016-67009-25639). The funders had no role in study design, data collection and analysis, decision to publish, or preparation of the manuscript.

==============================
Systems for collecting image data in conjunction with computer vision techniques are a powerful tool for increasing the temporal resolution at which plant phenotypes can be measured non-destructively. Computational tools that are flexible and extendable are needed to address the diversity of plant phenotyping problems. We previously described the Plant Computer Vision (PlantCV) software package, which is an image processing toolkit for plant phenotyping analysis. The goal of the PlantCV project is to develop a set of modular, reusable, and repurposable tools for plant image analysis that are open-source and community-developed. Here we present the details and rationale for major developments in the second major release of PlantCV. In addition to overall improvements in the organization of the PlantCV project, new functionality includes a set of new image processing and normalization tools, support for analyzing images that include multiple plants, leaf segmentation, landmark identification tools for morphometrics, and modules for machine learning.

Introduction

All approaches for improving crops eventually require measurement of traits (phenotyping) (Fahlgren, Gehan & Baxter, 2015). However, manual plant measurements are time-consuming and often require destruction of plant materials in the process, which prevents measurement of traits for a single plant through time. Consequently, plant phenotyping is widely recognized as a major bottleneck in crop improvement (Furbank & Tester, 2011). Targeted plant phenotypes can range from measurement of gene expression, to flowering time, to grain yield; therefore, the software and hardware tools used are often diverse. Here, we focus on the software tools required to nondestructively measure plant traits through images. This is a challenging area of research because the visual definition of phenotypes vary depending on the target species. For example, identification of petals can be used to measure flowering time, but petal color can vary by species. Therefore, software tools needed to process high-throughput image data need to be flexible and amenable to community input.

The term ‘high-throughput’ is relative to the difficulty to collect the measurement. The scale that might be considered high-throughput for root phenotyping might not be the same for shoot phenotyping, which can be technically easier to collect depending on the trait and species. Here we define high-throughput as thousands or hundreds of thousands of images per dataset. PlantCV is an open-source, open-development suite of analysis tools capable of analyzing high-throughput image-based phenotyping data (Fahlgren et al., 2015). Version 1.0 of PlantCV (PlantCV v1.0) was released in 2015 alongside the introduction of the Bellwether Phenotyping Facility at the Donald Danforth Plant Science Center (Fahlgren et al., 2015). PlantCV v1.0 was envisioned as a base suite of tools that the community could build upon, which lead to several design decisions aimed at encouraging participation. First, GitHub was used as a platform to organize the community by integrating version control, code distribution, documentation, issue tracking, and communication between users and contributors (Perez-Riverol et al., 2016). Second, PlantCV was written in Python, a high-level language widely used for both teaching and bioinformatics (Mangalam, 2002; Dudley & Butte, 2009), to facilitate contribution from both biologists and computer scientists. Additionally, the use of Python allows extension of PlantCV with the many tools available from the Python scientific computing community (Oliphant, 2007; Millman & Aivazis, 2011). Third, a focus on modular development fosters code reuse and makes it easier to integrate PlantCV with new or existing systems. Finally, the use of a permissive, open-source license (MIT) allows PlantCV to be used, reused, or repurposed with limited restrictions, for both academic and proprietary applications. The focus of the paper associated with the original release of PlantCV v1.0 (Fahlgren et al., 2015) was not the structure and function of PlantCV for image analysis, but rather an example of the type of biological question that can be answered with high-throughput phenotyping hardware and software platforms. Since the release of PlantCV v1.0 major improvements have been made to increase the flexibility, usability, and functionality of PlantCV, while maintaining all of the functionality in v1.0. Here we document the structure of PlantCV v2 along with examples that demonstrate new functionality.

Materials & Methods

The latest version or a specific release of PlantCV can be cloned from GitHub. The release for this paper is v2.1. Scripts, notebooks, SQL schema, and simple input data associated with the figures and results presented in this paper are available on GitHub at https://github.com/danforthcenter/plantcv-v2-paper. Project-specific GitHub repositories are kept separate from the PlantCV software repository because their purpose is to make project-specific analyses available for reproducibility, while the main PlantCV software repository contains general purpose image analysis modules, utilities, and documentation.

Images of Arabidopsis thaliana were captured with a Raspberry Pi computer and camera in a Conviron growth chamber. Additional details about the imaging set-up are provided in a companion paper (Tovar et al., 2017). Images of Setaria viridis (A10) and Setaria italica (B100) are from publicly available datasets that are available at http://plantcv.danforthcenter.org/pages/data.html (Fahlgren et al., 2015; Feldman et al., 2017). Images of wheat (Triticum aestivum L.) infected with wheat stem rust (Puccinia graminis f. sp. tritici) were acquired with a flatbed scanner.

Image analysis was done in PlantCV using Python v2.7.5, OpenCV v2.4.5 (Bradski, 2000), NumPy v1.12.1 (Van der Walt, Colbert & Varoquaux, 2011), Matplotlib v2.0.2 (Hunter, 2007), SciPy v0.19.0 (Jones, Oliphant & Peterson, 2014), Pandas v0.20.1 (McKinney, 2010), scikit-image v0.13.0 (Van der Walt et al., 2014), and Jupyter Notebook v4.2.1 (Kluyver et al., 2016). Statistical analysis and data visualization was done using R v3.3 (R Core Team, 2017) and RStudio v1.0 (RStudio Team, 2016). Graphs were produced using Matplotlib v2.0.2 (Hunter, 2007) and ggplot2 v2.2.1 (Wickham, 2009).

Results and Discussion

The following are details on improvements to the structure, usability, and functionality of PlantCV since the v1.0 release. Further documentation for using PlantCV can be found at the project website (http://plantcv.danforthcenter.org/).

Organization of the PlantCV project

PlantCV is a collection of modular Python functions, which are reusable units of Python code with defined inputs and outputs (Fig. 1A). PlantCV functions can be assembled into simple sequential or branching/merging pipelines. A pipeline can be as long or as short as it needs to be, allowing for maximum flexibility for users using different imaging systems and analyzing features of seed, shoot, root, or other plant systems. Suggestions on how to approach image analysis with PlantCV, in addition to specific tutorials, are available through online documentation (http://plantcv.readthedocs.io/en/latest/analysis_approach/). Each function has a debugging option to allow users to view and evaluate the output of a single step and adjust parameters as necessary. A PlantCV pipeline is written by the user as a Python script. Once a satisfactory pipeline script is developed, the PlantCV parallelization script (‘plantcv-pipeline.py’) can be used to deploy the pipeline across a large set of image data (Fig. 1A). The parallelization script also functions to manage data by consolidating measurements and metadata into an SQLite database (Fig. 1B). In terms of speed, the user is only limited by the complexity of the pipeline and the number of available processors.

Figure 1 Diagram of the components of PlantCV.

(A) PlantCV is an open-source, open-development suite of image analysis tools. PlantCV contains a library of modular Python functions that can be assembled into simple sequential or branching/merging processing pipelines. Image processing pipelines, which process single images (possibly containing multiple plants), can be deployed over large image sets using PlantCV parallelization, which outputs an SQLite database of both measurements and image/experimental metadata. (B) Overview of the structure of the SQLite database.

The modular structure of the PlantCV package makes it easier for members of the community to become contributors. Contributors to PlantCV submit bug reports, develop new functions and unit tests, or extend existing functionality or documentation. Core PlantCV developers do not filter additions of new functions in terms of perceived impact or number of users but do check that new functions follow the PlantCV contribution guide (see the sections on contributing in the online documentation). PlantCV contributors are asked to follow the PEP8 Python style guide (https://www.python.org/dev/peps/pep-0008/). Additions or revisions to the PlantCV code or documentation are submitted for review using pull requests via GitHub. The pull request mechanism is essential to protect against merge conflicts, which are sections of code that have been edited by multiple users in potentially incompatible ways.

In PlantCV v2, several service integrations were added to automate common tasks during pull requests and updates to the code repository. A continuous integration framework using the Travis CI service (https://travis-ci.org/) was added so that software builds and unit tests can be run automatically upon pull requests and other software updates. Continuous integration provides a safeguard against code updates that break existing functionality by providing a report that shows which tests passed or failed for each build (Wilson et al., 2014). The effectiveness of continuous integration depends on having thorough unit test coverage of the PlantCV code base. Unit test coverage of the PlantCV Python package is monitored through the Coveralls service (https://coveralls.io/), which provides a report on which parts of the code are covered by existing unit tests. In addition to the code, the PlantCV documentation was enhanced to use a continuous documentation framework using the Read the Docs service (https://readthedocs.org/), which allows documentation to be updated automatically and versioned in parallel with updates to PlantCV. The documentation was updated to cover all functions in the PlantCV library, tutorials on building pipelines and using specialized tools (e.g., multi-plant analysis and machine learning tools), a frequently asked questions section, and several guides such as installation, Jupyter notebooks, and instructions for contributors.

Improved usability

PlantCV v1.0 required pipeline development to be done using the command line, where debug mode is used to write intermediate image files to disk for each step. In command-line mode, an entire pipeline script must be executed, even if only a single step is being evaluated. To improve the pipeline and function development process in PlantCV v2, the debugging system was updated to allow for seamless integration with the Juptyer Notebook system (http://jupyter.org/; Kluyver et al., 2016). Jupyter compatibility allows users to immediately visualize output and to iteratively rerun single steps in a multi-step PlantCV pipeline, which makes parameters like thresholds or regions of interest much easier to adjust. Once a pipeline is developed in Jupyter, it can then be converted into a Python script that is compatible with PlantCV parallelization (see online documentation for detailed instructions on conversion; http://plantcv.readthedocs.io/en/latest/jupyter/). Because ofthe web-based interface and useful export options, Jupyter notebooks are also a convenient method of sharing pipelines with collaborators, or in publications, and teaching others to use PlantCV.

PlantCV was initially created to analyze data generated by the Bellwether Phenotyping Facility at the Donald Danforth Plant Science Center. Several updates to PlantCV v2 addressed the need to increase the flexibility of PlantCV to analyze data from other plant phenotyping systems. The PlantCV SQLite database schema was simplified so that new tables do not need to be added for every new camera system (Fig. 1B). The full database schema is available on GitHub (see ‘Materials and Methods’) and in PlantCV documentation. New utilities were added to PlantCV v2 that allow data to be quickly and efficiently exported from the SQLite database into text files that are compatible with R (R Core Team, 2017) for further statistical analysis and data visualization.

Because standards for data collection and management for plant phenotyping data are still being developed (Pauli et al., 2016), image metadata is often stored in a variety of formats on different systems. A common approach is to include metadata within image filenames, but because there is a lack of file naming standards, it can be difficult to robustly capture this data automatically. In PlantCV v2, a new metadata processing system was added to allow for flexibility in file naming both within and between experiments and systems. The PlantCV metadata processing system is part of the parallelization tool and works by using a user-provided template to process filenames. User-provided templates are built using a restricted vocabulary so that metadata can be collected in a standardized way. The vocabulary used can be easily updated to accommodate future community standards.

Performance

In PlantCV v1.0, image analysis parallelization was achieved using a Perl-based multi-threading system that was not thread-safe, which occasionally resulted in issues with data output that had to be manually corrected. Additionally, the use of the Python package Matplotlib (Hunter, 2007) in PlantCV v1.0 limited the number of usable processors to 10–12. For PlantCV v2, the parallelization framework was completely rewritten in Python using a multiprocessing framework, and the use of Matplotlib was updated to mitigate the issues and processor constraints in v1.0. The output of image files mainly used to assess image segmentation quality is now optional, which should generally increase computing performance. Furthermore, to decentralize the computational resources needed for parallel processing and prepare for future integration with high-throughput computing resources that use file-in-file-out operations, results from PlantCV pipeline scripts (one per image) are now written out to temporary files that are aggregated by the parallelization tool after all image processing is complete.

New functionality

PlantCV v2 has added new functions for image white balancing, auto-thresholding, size marker normalization, multi-plant detection, combined image processing, watershed segmentation, landmarking, and a trainable naive Bayes classifier for image segmentation (machine learning). The following are short descriptions and sample applications of new PlantCV functions.

White balancing

If images are captured in a greenhouse, growth chamber, or other situation where light intensity is variable, image segmentation based on global thresholding of image intensity values can become variable. To help mitigate image inconsistencies that might impair the ability to use a single global threshold and thus a single pipeline over a set of images, a white balance function was developed. If a white color standard is visible within the image, the user can specify a region of interest. If a specific area is not selected then the whole image is used. Each channel of the image is scaled relative to the reference maximum.

Auto-thresholding functions

An alternative approach to using a fixed, global threshold for image segmentation is to use an auto-thresholding technique that either automatically selects an optimal global threshold value or introduces a variable threshold for different regions in an image. Triangle, Otsu, mean, and Gaussian auto-thresholding functions were added to PlantCV to further improve object detection when image light sources are variable. The ‘triangle_auto_threshold’ function implements the method developed by Zack, Rogers & Latp (1977). The triangle threshold method uses the histogram of pixel intensities to differentiate the target object (plant) from background by generating a line from the peak pixel intensity (Duarte, 2015) to the last pixel value and then finding the point (i.e., the threshold value) on the histogram that maximizes distance to that line. In addition to producing the thresholded image in debug mode, the ‘triangle_auto_threshold’ function outputs the calculated threshold value and the histogram of pixel intensities that was used to calculate the threshold. In cases where the auto-threshold value does not adequately separate the target object from background, the threshold can be adjusted by modifying the stepwise input. Modifying the stepwise input shifts the distance calculation along the x-axis, which subsequently calculates a new threshold value to use.

The Otsu, mean, and Gaussian threshold functions in PlantCV are implemented using the OpenCV library (Bradski, 2000). Otsu’s binarization (‘otsu_auto_threshold;’ (Otsu, 1979)) is best implemented when a grayscale image histogram has two peaks since the Otsu method selects a threshold value that minimizes the weighted within-class variance. In other words, the Otsu method identifies the value between two peaks where the variances of both classes are minimized. Mean and Gaussian thresholding are executed by indicating the desired threshold type in the function ‘adaptive_threshold.’ The mean and Gaussian methods will produce a variable local threshold where the threshold value of a pixel location depends on the intensities of neighboring pixels. For mean adaptive thresholding, the threshold of a pixel location is calculated by the mean of surrounding pixel values; for Gaussian adaptive thresholding, the threshold value of a pixel is the weighted sum of neighborhood values using a Gaussian window (Gonzalez & Woods, 2002; Kaehler & Bradski, 2016).

Gaussian blur

In addition to the ‘median_blur’ function included in PlantCV v1.0, we have added a Gaussian blur smoothing function to reduce image noise and detail. Both the median and Gaussian blur methods are implemented using the OpenCV library (Bradski, 2000) and are typically used to smooth a grayscale image or a binary image that has been previously thresholded. Image blurring, while reducing detail, can help remove or reduce signal from background noise (e.g., edges in imaging cabinets), generally with minimal impact on larger structures of interest. Utilizing a rectangular neighborhood around a center pixel, ‘median_blur’ replaces each pixel in the neighborhood with the median value. Alternatively, ‘gaussian_blur’ determines the value of the central pixel by multiplying its and neighboring pixel values by a normalized kernel and then averaging these weighted values (i.e., image convolution) (Kaehler & Bradski, 2016). The extent of image blurring can be modified by increasing (for greater blur) or decreasing the kernel size (which takes only odd numbers; commonly, 3 × 3) or by changing the standard deviation in the X and/or Y directions.

Size marker normalization

Images that are not collected from a consistent vantage point require one or more size markers as references for absolute or relative scale. The size marker function allows users to either detect a size marker within a user-defined region of interest or to select a specific region of interest to use as the size marker. The pixel area of the marker is returned as a value that can be used to normalize measurements to the same scale. For this module to function correctly we assume that the size marker stays in frame, is unobstructed, and is relatively consistent in position throughout a dataset, though some movement is allowed as long as the marker remains within the defined marker region of interest.

Multi-plant detection

There is growing interest among the PlantCV user community to process images with multiple plants grown in flats or trays, but PlantCV v1.0 was built to processes images containing single plants. The major challenge with analyzing multiple plants in an image is successfully identifying individual whole plants as distinct objects. Leaves or other plant parts can sometimes be detected as distinct contours from the rest of the plant and need to be grouped with other contours from the same plant to correctly form a single plant/target object. While creating multiple regions of interest (ROI) to demarcate each area containing an individual plant/target is an option, we developed two modules, ‘cluster_contours’ and ‘cluster_contours_split_img,’ that allow contours to be clustered and then parsed into multiple images without having to manually create multiple ROIs (Fig. 2).

Figure 2 Analysis of images containing multiple plants.

New functions have been added to PlantCV v2 that enable individual plants from images containing multiple plants to be analyzed. The ‘cluster_contours’ function clusters contour objects using a flexible grid arrangement (approximate rows and columns defined by a user). (A) An image produced by ‘cluster_contours’ in debug mode highlights plants by their cluster group with unique colors on a sequential scale. The ‘cluster_contours_split_img’ function creates a new image for each cluster group. The resulting images of individual plants can be processed by standard PlantCV methods. (B) The ‘cluster_contours_split_img’ function was used to split the full image into individual plants. The shape of each plant was then analyzed with ‘analyze_objects’ and printed on a common image background.

The ‘cluster_contours’ function takes as input: an image, the contours that need to be clustered, a number of rows, and a number of columns. Total image size is detected, and the rows and columns create a grid to serve as approximate ROIs to cluster the contours (Fig. 2A). The number of rows and columns approximate the desired size of the grid cells. There does not need to be an object in each of the grid cells. Several functions were also added to aid the clustering function. The ‘rotate_img’ and ‘shift_img’ functions allow the image to be adjusted so objects are better aligned to a grid pattern.

After objects are clustered, the ‘cluster_contour_split_img’ function splits images into the individual grid cells and outputs each as a new image so that there is a single clustered object per image. If there is no clustered object in a grid cell, no image is outputted. With the ‘cluster_contour_split_img’ function, a text file with genotype names can be included to add them to image names. The ‘cluster_contour_split_img’ function also checks that there are the same number of names as objects. If there is a conflict in the number of names and objects, a warning is printed and a correction is attempted. Alternatively, if the file option is not used, all of the object groups are labeled by position. Once images are split, they can be processed like single plant images using additional PlantCV tools (Fig. 2B). See the online documentation for an example multi-plant imaging pipeline (http://plantcv.readthedocs.io/en/latest/multi-plant_tutorial/).

The current method for multi-plant identification in PlantCV is flexible but relies on a grid arrangement of plants, which is common for controlled-environment-grown plants. Future releases of PlantCV may incorporate additional strategies for detection and identification of plants, such as arrangement-independent K-means clustering approaches (Minervini, Abdelsamea & Tsaftaris, 2014).

Combined image processing

The Bellwether Phenotyping Facility has both RGB visible light (VIS) and near-infrared (NIR) cameras, and images are captured ∼1 min apart (Fahlgren et al., 2015). Compared to VIS images, NIR images are grayscale with much less contrast between object and background. It can be difficult to segment plant material from NIR images directly, even with edge detection steps. Therefore, several functions were added to allow the plant binary mask that results from VIS image processing pipelines to be resized and used as a mask for NIR images. Combining VIS and NIR camera pipelines also has the added benefit of decreasing the number of steps necessary to process images from both camera types, thus increasing image processing throughput. The ‘get_nir’ function identifies the path of the NIR image that matches VIS image. The ‘get_nir’ function requires that the image naming scheme is consistent and that the matching image is in the same image directory. The ‘resize’ function then resizes the VIS plant mask in both the x and y directions to match the size of the NIR image. Resizing values are determined by measuring the same reference object in an example image taken from both VIS and NIR cameras (for example the width of the pot or pot carrier in each image). The ‘crop_position_mask’ function is then used to adjust the placement of the VIS mask over the NIR image and to crop/adjust the VIS mask so it is the same size as the NIR image. It is assumed that the pot position changes consistently between VIS and NIR image datasets. An example VIS/NIR dual pipeline to follow can be accessed online (http://plantcv.readthedocs.io/en/latest/vis_nir_tutorial/).

Figure 3 Leaf segmentation by a distance-based watershed transformation.

The watershed segmentation function can be used to segment and estimate the number of objects in an image. For the three example images, the watershed segmentation function was used to estimate the number of leaves for Arabidopsis thaliana (estimated leaf count for top: 13, middle: 14, and bottom: eight). Images shown are the output from the ‘watershed_segmentation’ function (A, C, E) and the segmented plants (B, D, F).

Object count estimation with watershed segmentation

While segmentation and analysis of whole plants in images provides useful information about plant size and growth, a more detailed understanding of plant growth and development can be obtained by measuring individual plant organs. However, fully automated segmentation of individual organs such as leaves remains a challenge, due to issues such as occlusion (Scharr et al., 2016). Multiple methods for leaf segmentation have been proposed (Scharr et al., 2016), and in PlantCV v2 we have implemented a watershed segmentation approach. The ‘watershed_segmentation’ function can be used to estimate the number of leaves for plants where leaves are distinctly separate from other plant structures (e.g., A. thaliana leaves are separated by thin petioles; Fig. 3). The inputs required are an image, an object mask, and a minimum distance to separate object peaks. The function uses the input mask to calculate a Euclidean distance map (Liberti et al., 2014). Marker peaks calculated from the distance map that meet the minimum distance setting are used in a watershed segmentation algorithm (Van der Walt et al., 2014) to segment and count the objects. Segmented objects are visualized in different colors, and the number of segmented objects is reported (Fig. 3). An example of how the watershed segmentation method was used to assess the effect of water deficit stress on the number of leaves of A. thaliana plants can be found in Acosta-Gamboa et al. (2017).

Landmarking functions for morphometrics

To extend PlantCV beyond quantification of size-based morphometric features, we developed several landmarking functions. Landmarks are generally geometric points located along the contours of a shape that correspond to homologous biological features that can be compared between subjects (Bookstein, 1991). Typical examples of landmarks include eyes between human subjects or suture joins in a skull. For a growing plant, potential landmarks include the tips of leaves and pedicel and branch angles. When specified a priori, landmarks should be assigned to provide adequate coverage of the shape morphology across a single dimensional plane (Bookstein, 1991). Additionally, the identification of landmark points should be repeatable and reliable across subjects while not altering their topological positions relative to other landmark positions (Bookstein, 1991). Type I landmarks provide the strongest support for homology because they are defined by underlying biological features, but it is problematic to assign Type I landmarks a priori when analyzing high-throughput plant imagery. To address this, PlantCV v2 contains functions to identify anatomical landmarks based upon the mathematical properties of object contours (Type II) and non-anatomical pseudo-landmarks/semilandmarks (Type III), as well as functions to rescale and analyze biologically relevant shape properties (Bookstein, 1991; Bookstein, 1997; Gunz, Mitteroecker & Bookstein, 2005; Gunz & Mitteroecker, 2013).

The ‘acute’ function identifies Type II landmarks by implementing a pseudo-landmark identification algorithm that operates using a modified form of chain coding (Freeman, 1961). Unlike standard chain coding methods that attempt to capture the absolute shape of a contour, the acute method operates by measuring the angle between a pixel coordinate and two neighboring pixels on opposite sides of it that fall within a set distance, or window, along the length of the contour. The two neighboring points are used to calculate an angle score for the center pixel. When the angle score is calculated for each position along the length of a contour, clusters of acute points can be identified, which can be segmented out by applying an angle threshold. The middle position within each cluster of acute points is then identified for use as a pseudo-landmark (Fig. 4A). The ability to subjectively adjust the window size used for generating angle scores also helps to tailor analyses for identifying points of interest that may differ in resolution. For example, an analysis of leaf data might utilize a larger window size to identify the tips of lobes whereas smaller window sizes would be able to capture more minute patterns such as individual leaf serrations. Further segmentation can also be done using the average pixel values output (pt_vals) for each pseudo-landmark, which estimates the mean pixel intensity within the convex hull of each acute region based on the binary mask used in the analysis. The average pixel value output allows for concave landmarks (e.g., leaf axils and grass ligules) and convex landmarks (e.g., leaf tips and apices) on a contour to be differentiated in downstream analyses. Additionally, PlantCV v2 includes the ‘acute_vertex’ function that uses the same chain code-based pseudo-landmark identification algorithm used in the ‘acute’ function except that it uses an adjustable local search space criteria to reduce the number of angle calculations, which speeds up landmark identification.

Figure 4 Landmark-based analysis of plant shape in PlantCV.

(A) Automatic identification of leaf tip landmarks using the ‘acute’ and ‘acute_vertex’ functions (blue dots). (B) Geometrically homologous semi/pseudo-landmarks across both the x- and y-axes. Semi/pseudo-landmarks identified by scanning the x-axis are denoted by light blue (top side of the contour), brown (bottom side of the contour), and light orange (centroid location of horizontal bins) dots. Semi/pseudo-landmarks identified by scanning the y-axis are denoted by dark blue (left side of the contour), pink (right side of the contour), and dark orange (centroid location of vertical bins) dots. The plant centroid is plotted larger in red. (C) A representation of the rescaled plant landmarks identified in panel (A). White points correspond to the leaf tips. The orange point is the location of the plant centroid. The blue point is the location of the plant centroid where the plant emerges from the soil. Red lines are the vertical distance from leaf tip points relative to the plant centroid. (D) Analysis of the average scaled vertical distance from each leaf tip to the centroid diverges in response to water limitation.

For Type III landmarks, the ‘x_axis_pseudolandmarks’ and ‘y_axis_pseudolandmarks’ functions identify homologous points along a single dimension of an object (x-axis or y-axis) based on equidistant point locations within an object contour. The plant object is divided up into twenty equidistant bins, and the minimum and maximum extent of the object along the axis and the centroid of the object within each bin is calculated. These sixty points located along each axis possess the properties of semi/pseudo-landmark points (an equal number of reference points that are approximately geometrically homologous between subjects to be compared) that approximate the contour and shape of the object (Fig. 4B). Such semi/pseudo-landmarking strategies have been utilized in cases where traditional homologous landmark points are difficult to assign or poorly represent the features of object shape (Bookstein, 1997; Gunz, Mitteroecker & Bookstein, 2005; Gunz & Mitteroecker, 2013).

Frequently, comparison of shape attributes requires rescaling of landmark points to eliminate the influence of size on the relative position of landmark points. The landmark functions in PlantCV output untransformed point values that can either be directly input into morphometric programs in R (shapes (Dryden & Mardia, 2016) or morpho (Schlager, 2017)) or uniformly rescaled to a 0-1 coordinate system using the PlantCV ‘scale_features’ function. The location of landmark points can be used to examine multidimensional growth curves for a broad variety of study systems and tissue types and can be used to compare properties of plant shape throughout development or in response to differences in plant growth environment. An example of one such application is the ‘landmark_reference_pt_dist’ function. This function estimates the vertical, horizontal, Euclidean distance, and angle of landmark points from two landmarks (centroid of the plant object and centroid localized to the base of the plant). Preliminary evidence from a water limitation experiment performed using a Setaria recombinant inbred population indicates that vertical distance from rescaled leaf tip points identified by the ‘acute_vertex’ function to the centroid is decreased in response to water limitation and thus may provide a proximity measurement of plant turgor pressure (Figs. 4C and 4D).

Two-class or multiclass naive Bayes classifier

Pixel-level segmentation of images into two or more classes is not always straightforward using traditional image processing techniques. For example, two classes of features in an image may be visually distinct but similar enough in color that simple thresholding is not sufficient to separate the two groups. Furthermore, even with methods that adjust for inconsistencies between images (e.g., white balancing and auto-thresholding functions), inconsistent lighting conditions in a growth chamber, greenhouse, or field can still make bulk processing of images with a single workflow difficult. Methods that utilize machine learning techniques are a promising approach to tackle these and other phenotyping challenges (Minervini, Abdelsamea & Tsaftaris, 2014; Singh et al., 2016; Ubbens & Stavness, 2017; Atkinson et al., 2017; Pound et al., 2017). With PlantCV v2, we have started to integrate machine learning methods to detect features of interest (e.g., the plant), starting with a naive Bayes classifier (Abbasi & Fahlgren, 2016). The naive Bayes classifier can be trained using two different approaches for two-class or multiclass (two or more) segmentation problems. During the training phase using the ‘plantcv-train.py’ script, pixel RGB values for each input class are converted to the hue, saturation and value (HSV) color space. Kernel density estimation (KDE) is used to calculate a probability density function (PDF) from a vector of values for each HSV channel from each class. The output PDFs are used to parameterize the naive Bayes classifier function (‘naive_bayes_classifier’), which can be used to replace the thresholding steps in a PlantCV pipeline. The ‘naive_bayes_classifer’ function uses these PDFs to calculate the probability (using Bayes’ theorem) that a given pixel is in each class. The output of the ‘naive_bayes_classifier’ is a binary image for each class where the pixels are white if the probability the pixel was in the given class was highest of all classes and is black otherwise. A tutorial of how to implement naive Bayes plant detection into an image processing pipeline is online (http://plantcv.readthedocs.io/en/latest/machine_learning_tutorial/).

For the two-class approach, the training dataset includes color images and corresponding binary masks where the background is black and the foreground (plant or other target object) is white. PlantCV can be used to generate binary masks for the training set using the standard image processing methods and the new ‘output_mask’ function. It is important for the training dataset to be representative of the larger dataset. For example, if there are large fluctuations in light intensity throughout the day or plant color throughout the experiment, the training dataset should try to cover the range of variation. A random sample of 10% of the foreground pixels and the same number background pixels are used to build the PDFs.

To assess how well the two-class naive Bayes method identifies plant material in comparison to thresholding methods, we reanalyzed Setaria images (Fahlgren et al., 2015) using the naive Bayes classifier and compared the pixel area output to pipelines that utilize thresholding steps (Fig. 5). We used 99 training images (14 top view and 85 side view images) from a total of 6,473 images. We found that the plant pixel area calculated by naive Bayes was highly correlated with that calculated from pipelines that use thresholding for both side-view images (R2 = 0.99; Fig. 5A) and top-view images (R2 = 0.96; Fig. 5B). Naive Bayes segmentation enabled use of pipelines that were both simpler (fewer steps) and more flexible: five new scripts were sufficient for processing the dataset (five categories of photo data), whereas nine threshold-based pipeline scripts had previously been required.

Figure 5 Plant segmentation using a naive Bayes classifier.

Correlation between plant area in pixels (px) detected using thresholding pipelines (Fahlgren et al., 2015) on the x-axis compared to plant area detected using a trained naive Bayes classifier on the y-axis. (A) Side-view images. (B) Top-view images.

The multiclass naive Bayes approach requires a tab-delimited table for training where each column is a class (minimum two) and each cell is a comma-separated list of RGB pixel values from the column class. We currently use the Pixel Inspection Tool in ImageJ (Schneider, Rasband & Eliceiri, 2012) to collect samples of pixel RGB values used to generate the training text file. As noted above for the two-class approach, it is important to adequately capture the variation in the image dataset for each class when generating the training text file to improve pixel classification. If images are consistent, only one image needs to be sampled for generating the training table; however, if they vary, several images may be needed. For complex backgrounds (or non-targeted objects), several classes may be required to capture all of the variation. Once the training table is generated, it is input into the ‘plantcv-train.py’ script to generate PDFs for each class. As an example, we used images of wheat leaves infected with wheat rust to collect pixel samples from four classes: non-plant background, unaffected leaf tissue, rust pustule, and chlorotic leaf tissue, and then used the naive Bayes classifier to segment the images into each class simultaneously (Fig. 6). This method can likely be used for a variety of applications, such as identifying a plant under variable lighting conditions or quantifying specific areas of stress on a plant.

Figure 6 Simultaneous segmentation of four feature groups using the naive Bayes classifier.

An example of the naive Bayes classifier used to assign pixels into four classes: background, unaffected plant tissue, chlorotic tissue, and wheat stem rust pustules. (A) Probability density functions (PDFs) from the ‘plantcv-train.py’ script that show hue, saturation, and value color channel distributions of four classes estimated from training data. (B) Example of a classified image. Photo credit: Katie Liberatore and Shahryar Kianian. (C) Example of a merged pseudocolored image with pixels classified by the ‘naive_bayes_classifier’ as background (black), unaffected leaf tissue (green), chlorotic leaf tissue (blue), and pustules (red).

In summary, the naive Bayes classifier offers several advantages over threshold-based segmentation: (1) two or more classes can be segmented simultaneously; (2) probabilistic segmentation can be more robust across images than fixed thresholds; and (3) classifier-based segmentation replaces multiple steps in threshold-based pipelines, reducing pipeline complexity.

Conclusions

The field of digital plant phenotyping is at an exciting stage of development where it is beginning to shift from a bottleneck to one that will have a positive impact on plant research, especially in agriculture. The Plant Image Analysis database currently lists over 150 tools that can be used for plant phenotyping (http://www.plant-image-analysis.org/; Lobet, Draye & Périlleux, 2013). Despite the abundance of software packages, long-term sustainability of individual projects may become an issue due to the lack of incentives for maintaining bioinformatics software developed in academia (Lobet, 2017). In a survey of corresponding authors of plant image analysis tools by Lobet, 60% either said the tool was no longer being maintained or did not respond (Lobet, 2017). To develop PlantCV as a sustainable project we have adopted an open, community-based development framework using GitHub as a central service for the organization of developer activities and the dissemination of information to users. We encourage contribution to the project by posting bug reports and issues, developing or revising analysis methods, adding or updating unit tests, writing documentation, and posting ideas for new features. We aim to periodically publish updates, such as the work presented here, to highlight the work of contributors to the PlantCV project.

There are several areas where we envision future PlantCV development. Standards and interoperability: Improved interoperability of PlantCV with data providers and downstream analysis tools will require adoption of community-based standards for data and metadata (e.g., Minimum Information About a Plant Phenotyping Experiment; Ćwiek Kupczyńska et al., 2016). Improved interoperability will make it easier to develop standardized tools for statistical analysis of image processing results, both within the PlantCV project or with tools from other projects. New data sources: Handling and analysis of data from specialized cameras that measure three-dimensional structure or hyperspectral reflectance will require development or integration of additional methods into PlantCV. Machine learning: Our goal is to develop additional tools for machine learning and collection of training data. In some cases, where these methods can be implemented in a modular and reusable framework, they can be integrated directly into PlantCV. In other cases, PlantCV can be combined with new and existing tools. A recent example of this latter approach built on PlantCV, using its image preprocessing and segmentation functions alongside a modular framework for building convolutional neural networks (Ubbens & Stavness, 2017). As noted throughout, we see great potential for modular tools such as PlantCV and we welcome community feedback.

We would like to thank Melinda Darnell, Leonardo Chavez, Kevin Reilly, and the staff of both the Danforth Center Facilities and Support Services group and the Plant Growth Facility for careful maintenance of the Danforth Center phenotyping facilities. We thank Katie Liberatore and Shahryar Kianian for images of wheat (Triticum aestivum L.). We would also like to thank all of the other people who have given us input on the PlantCV project in person or on GitHub.

Additional Information and Declarations

Competing Interests

Author Contributions

Data Availability

Malia A. Gehan, Noah Fahlgren, Arash Abbasi, Jeffrey C. Berry, Steven T. Callen, Leonardo Chavez, Max J. Feldman, Kerrigan B. Gilbert, Steen Hoyer, Andy Lin, César Lizárraga, Michael Miller and Monica Tessman contributed to the research described while working at the Donald Danforth Plant Science Center, a 501(c)(3) nonprofit research institute. Suxing Liu and Argelia Lorence contributed to the research described while working at the University of Arkansas. John G. Hodge and Andrew N. Doust contributed to the research described while working at the University of Oklahoma. Eric Platon contributed to the research described while working as a founder and employee of Cosmos X. Tony Sax contributed to the research described while a full-time student at the Missouri University of Science and Technology.

Malia A. Gehan, Noah Fahlgren and Max J. Feldman conceived and designed the experiments, performed the experiments, analyzed the data, contributed reagents/materials/analysis tools, wrote the paper, prepared figures and/or tables, reviewed drafts of the paper.

Arash Abbasi, Andrew N. Doust and John G. Hodge contributed reagents/materials/analysis tools, wrote the paper, reviewed drafts of the paper.

Jeffrey C. Berry, Leonardo Chavez, Andy Lin, César Lizárraga, Michael Miller, Eric Platon, Monica Tessman and Tony Sax contributed reagents/materials/analysis tools, reviewed drafts of the paper.

Steven T. Callen analyzed the data, contributed reagents/materials/analysis tools, wrote the paper, reviewed drafts of the paper.

Kerrigan B. Gilbert prepared figures and/or tables, reviewed drafts of the paper.

J. Steen Hoyer performed the experiments, contributed reagents/materials/analysis tools, wrote the paper, reviewed drafts of the paper.

Suxing Liu and Argelia Lorence conceived and designed the experiments, contributed reagents/materials/analysis tools, wrote the paper, reviewed drafts of the paper.

The following information was supplied regarding data availability:

PlantCV is available on GitHub at https://github.com/danforthcenter/plantcv. PlantCV v2.1 is archived on Zenodo at https://doi.org/10.5281/zenodo.1035894. Scripts used for image and statistical analysis are available on GitHub at https://github.com/danforthcenter/plantcv-v2-paper.

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
