# Peer review of "PlantCV v2: Image analysis software for high-throughput plant phenotyping"

_PeerJ, doi:10.7717/peerj.4088_

## Round 0.1 · original submission · Minor Revisions

Regarding the comment from R2 about providing the SQL schema:
I also had some questions about that. Does the repo at https://github.com/danforthcenter/plantcv contain this file? How was the .sql file in https://github.com/danforthcenter/plantcv-v2-paper generated? More generally, it was a bit unclear exactly how the paper repository coordinates with the main plantcv repository. Can you add some additional documentation to https://github.com/danforthcenter/plantcv-v2-paper to explain this? Lastly, I would recommend tagging the PlantCV repo commit that corresponds to the version discussed in the paper.

I enjoyed reading the article and look forward to the next steps!

Best wishes,
Ann

Reviewer 1 ·

Basic reporting

Nothing to add

Experimental design

The paper includes details of the listing of the available methods through PlantCV. This is fitting as the paper is about this new software.

The main impact of the paper would be enabling the potential users to leverage this software. In that regard, adding a few "how to" could be helpful. For example, what is the general guideline of which method to use for segmenting? And how to tune many parameters? How to create a series of steps to work for some common tasks? How to fix any errors from the algorithm? What is the expected speed of each method?

Validity of the findings

Nothing to add

Additional comments

The paper is a good introduction to the PlantCV software. Overall the paper and proposed method seem to address much need of the field. The future steps are insightful and should further increase impact. Main comments are to maximize the utilization of the tool by answering typical struggle that non-technical users would face.

·

Basic reporting

Language used is clear throughout and the relevant phenotyping literature is well cited. One of the sections I am direct to comment on here is whether the results are relevant to the hypothesis proposed, however this is a methods/resource paper and does not include hypothesis driven research.

Experimental design

No comment.

Validity of the findings

No comment.

Additional comments

Lines 52-54: The point about floral structures being different colors in different plants (and the need for fixable analysis approaches as a result) is a good one, but the specific example used opens up a big can of worms. The floral organs (ie flowers) of Camelina are yellow and the inflorescence/glumes of brachypodium are green. The floral organs of brachypodium (such as anthers) would also be yellow and since anthesis is often used as the milestone for "flowering" in grasses, it would often be the case that folks would be looking for yellow in photos of grass inflorescences. What about yellow Camelina flowers vs white arabidopsis flowers?

Line 85-86: I am afraid it wasn't clear what picture "The image of Arabidopsis thaliana" was in context here as there are several Arabidopsis images in different figures. Maybe include "used in Figure X or analysis Y" here. and for the wheat image listed a little later in this section.

In the discussion of unit tests it is not clear if the development and maintence of unit tests is an expectation of contributors who write new modules for incorporation into Plant CV or of the core management team.

155-158: I would encourage the authors to include a copy of the SQLite database schema as it exists with the writing of this paper as a figure or supplementary figure. I assume the schema on github will continue to be updates at PlantCV matures, which may make the reference to github less relevant to the version of the software described in this manuscript. This could potentially replace the current Figure 1.

Page 16: Could the authors add one to two sentences on the circumstances where a researcher might want to trade a reduction in image detail for a reduction in image noise?

Page 19: "The ‘watershed_segmentation’ function can be used to estimate the number of leaves for certain plant architecture types and imaging platforms (Fig. 3)" It might be useful to add a little more detail on the characteristics of plant architectures where the watershed method is likely to work or likely to fail at this point.

Pages 23-24: If the Bayesian method provides approximately the same output as a theshold-based analysis, but the Bayesian method requires fewer total steps that is a clear advantage to the Bayesian method. However, it wasn't clear what the other advantage(s) were to the Bayesian method since the sentence describing that point starts with "An additional advantage to using the naive Bayes..." (Lines 421-423).

Reviewer 3 ·

Basic reporting

Experimental design

Validity of the findings

Annotated reviews are not available for download in order to protect the identity of reviewers who chose to remain anonymous.

---

## Round 0.2 · accepted · Accept

Thank you for your revisions.

Congratulations and thank you for publishing with PeerJ.